# Polymerized Whey Protein Concentrate-Based Glutathione Delivery System: Physicochemical Characterization, Bioavailability and Sub-Chronic Toxicity Evaluation

**DOI:** 10.3390/molecules26071824

**Published:** 2021-03-24

**Authors:** Siyu Zhang, Cuina Wang, Weigang Zhong, Alyssa H. Kemp, Mingruo Guo, Adam Killpartrick

**Affiliations:** 1Key Laboratory of Dairy Science of Ministry of Education, Northeast Agricultural University, Harbin 150030, China; Zhangsiyu@neau.edu.cn (S.Z.); Ming.R.Guo@uvm.edu (M.G.); 2Department of Food Science, Jilin University, Changchun 130000, China; wangcuina@jlu.edu.cn (C.W.); zhongwg19@mails.jlu.edu.cn (W.Z.); 3Department of Nutrition and Food Sciences, University of Vermont, Burlington, VT 05403, USA; Alyssa.H.Kemp@uvm.edu

**Keywords:** polymerized whey protein, glutathione, physicochemical properties, pharmacokinetics, toxicity

## Abstract

Glutathione (GSH) is a powerful antioxidant, but its application is limited due to poor storage stability and low bioavailability. A novel nutrient encapsulation and delivery system, consisting of polymerized whey protein concentrate and GSH, was prepared and in vivo bioavailability, antioxidant capacity and toxicity were evaluated. Polymerized whey protein concentrate encapsulated GSH (PWPC-GSH) showed a diameter of roughly 1115 ± 7.07 nm (D_50_) and zeta potential of 30.37 ± 0.75 mV. Differential scanning calorimetry (DSC) confirmed that GSH was successfully dispersed in PWPC particles. In vivo pharmacokinetics study suggested that PWPC-GSH displayed 2.5-times and 2.6-fold enhancement in maximum concentration (C_max_) and area under the concentration–time curve (AUC) as compared to free GSH. Additionally, compared with plasma of mice gavage with free GSH, significantly increased antioxidant capacity of plasma in mice with PWPC-GSH was observed (*p* < 0.05). Sub-chronic toxicity evaluation indicated that no adverse toxicological reactions related to oral administration of PWPC-GSH were observed on male and female rats with a diet containing PWPC-GSH up to 4% (*w*/*w*). Data indicated that PWPC may be an effective carrier for GSH to improve bioavailability and antioxidant capacity.

## 1. Introduction

Glutathione (γ-glutamyl-cysteinyl-glycine, GSH) is an abundant endogenous antioxidant composed of glutamate, glycine and cysteine and mainly exists in baker’s yeast, wheat germ and animal liver. GSH is known to possess strong antioxidant properties as a result of its thiol (-SH) group inherent to the cysteine residue. The thiol group is involved in scavenging free radicals, thus protecting the integrity of the cell [1,2] and improving the immune system [3,4]. GSH can be a substrate of glutathione peroxidase (GSH-Px) and glutathione *S*-transferase (GST) [5], which have the function of detoxification [6]. GSH deficiency may play a role in various diseases, such as immunodeficiency disease [7], cancer [8], diabetes [9,10], cardiovascular disease [11,12], rheumatoid arthritis [13], and neurological disorders [14]. GSH has been used in clinical practice and consumer goods including cosmetic and function foods [15,16]. However, the poor storage stability, low bioavailability and ongoing debate as to whether orally administered GSH is absorbed still poses challenges for its broad clinical application. GSH is sensitive to temperature, light and pH due to the presence of the reduced thiol group. When exposed to external environment, the thiol group of GSH can be oxidized into a disulfide bond, forming oxidizing glutathione (GSSG), which reduces the antioxidant activity. In addition, GSH is easily degraded in intestinal tract and liver by γ-glutamyltransferase [17] and only a small amount is known to reach portal circulation. 

Whey protein, a by-product from the cheese industry, has two products available on market, which are whey protein concentrate (WPC) and whey protein isolate (WPI). Whey protein has high nutritional value and a complete amino acid profile, and has functions such as anti-oxidation [18,19], regulating immunity and inhibiting pathogenic bacteria. 

Whey protein can be used as carrier of delivery systems due to different mechanisms. Whey protein belongs to the lipocalin protein family and can be used to carry hydrophobic compounds such as polyphenols [20,21] mainly by hydrophobic forces. Whey protein has excellent emulsification properties and can be used to carry fat-soluble substances such as oil [22,23] and vitamins [24,25,26]. Polymerized whey protein has high viscosity and can be used to carry bioactive substances such as 3,3′-diindolylmethane [27] and probiotics [28]. 

Whey protein consists of a group of globular proteins including β-lactoglobulin, α-lactalbumin, and bovine serum albumin, which are sensitive to heat treatment. When heated, its spatial structure changes, resulting in the exposure of thiol and hydrophobic groups and the subsequent formation of polymerized whey protein via disulfide bond interconnections and hydrophobic interactions [29]. 

In this study, polymerized whey protein concentrate (PWPC), induced by thermal treatment of whey protein concentrate, was used to construct a new encapsulate and delivery system for GSH. To this end, physicochemical properties of the developed system were characterized followed by evaluation of bioavailability and sub-chronic toxicity.

## 2. Results and Discussion

### 2.1. Physicochemical and Structural Properties of the PWPC-GSH System

PWPC-GSH system was prepared with advantages of simplicity, mildness, and organic solvent-free in comparison with other carriers based on poly isobutylcyanoacrylate [30], Eudragit RS 100/cyclodextrin [31] and montmorillonite [32]. As shown in Figure 1A, PWPC exhibited bimodal pattern with two peaks at 594 nm and 4580 nm with a wide particle size distribution (span of 9.22), consistent with previous research [33]. Combination with GSH (287.83 ± 6.18 nm) slightly increased particle size (D_50_) from 1085 ± 35.35 to 1115 ± 7.07 nm with decreased span from 9.22 ± 0.22 to 6.86 ± 0.19. Zeta potential for PWPC-GSH was found to be 30.37 ± 0.75 mV (Figure 1B). The high surface charge endowed the PWPC-GSH system high stability since strong electrostatic repulsion between molecules prevents polymerization, precipitation, and flocculation [34]. In addition, the positive surface charge of PWPC-GSH would favor absorption in vivo since cell membranes carries negative charges [35]. PWPC-GSH system displayed shear-thinning behavior in range of 1–300 s^−1^ (Figure 1C), indicating that interaction between droplets was weakened at higher shear rate [36].

A DSC thermogram of GSH demonstrated an exothermic peak at 198 °C and disappearance of this melting peak in the PWPC-GSH system, implying that GSH was molecularly dispersed in PWPC particles (Figure 2A) [37]. FTIR spectra analysis (Figure 2B) showed that PWPC displayed an amide Ι (C=O vibration) spectrum peak at 1654.39 cm^−1^ and red shift occurred after binding with GSH, indicating that PWPC was structurally changed and intermolecular hydrogen bonds formed. This PWPC-GSH system exhibited morphology of vermicular aggregates with its majority at a size of roughly 200 nm, with some larger aggregates measuring upwards of approximately 400 nm (Figure 2C).

### 2.2. In Vivo Pharmacokinetic and Antioxidant Activity of PWPC-GSH 

Whey protein has been widely studied as an effective means of nutrient delivery due to its resistance to digestion by pepsin [38], its non-toxic nature, widely available sources and broad biocompatibility. Pharmacokinetic studies of the PWPC-GSH delivery system and free GSH were conducted and plasma GSH concentration–time profiles for all groups are shown in Figure 3A. GSH concentration was observed to be the highest in the plasma of PWPC-GSH group, followed by free GSH, PWPC, and the control group. The higher value in the plasma of mice gavage with PWPC-GSH than that of free GSH group may be due to the protection effect of highly viscous PWPC [39,40] by embedding GSH inside and preventing damage to gastrointestinal enzymes and an acidic environment. These results were consistent with previous studies that the bioavailability of quercetin and vitamin D were improved through whey protein encapsulating [41,42]. 

Pharmacokinetic parameters were calculated as shown in inset of Figure 3A. Compared with free GSH (maximum concentration (C_max_) of 7.37 mg/L and area under the concentration–time curve (AUC) of 19.23 h × mg/L), higher C_max_ (19.41 mg/L) and AUC (48.63 h × mg/L) values were observed, indicating a higher rate and degree of GSH absorption into the blood circulation in mice after administration with PWP-GSH. The 2.5-fold and 2.6-fold higher C_max_ and AUC in the PWPC-GSH group suggested that the PWPC-GSH delivery system can improve the in vivo bioavailability of GSH effectively vs. GSH in its pure form on its own. Whey protein also appears to possess a protective effect on GSH as a carrier during absorption into intestinal tract which may due to the resistance to digestion by pepsin. In addition to delivery of the GSH itself, the whey protein supplementation may have also contributed to the increase in GSH levels in vivo [43,44] by virtue of the abundance of cysteine residue inherent to whey protein, which has the capability to promote biosynthesis of GSH as a rate-limiting amino acid [45]. The lower time to maximum concentration (T_max_) (1 h) occurred in the PWPC-GSH group in comparison with that in free GSH (2 h), indicating less time was required to reach the maximum concentration after administration. The plasma concentration of GSH in the GSH group reached its maximum levels after 1.5 to 2 h which echoed data reported in the early literature relative to orally administered free GSH [46]. 

Total antioxidative capacity of samples at different time points was measured using an assay kit and the results are shown in Figure 3B. Antioxidant capacity of plasma in mice gavage with PWPC-GSH was significantly higher than that of free GSH through the whole period (*p* < 0.05). The first reason for the increased antioxidant capacity of plasma after gavage of PWPC-GSH in mice was that GSH concentration in plasma was improved using PWPC as a delivery carrier. The second reason may be due to the antioxidant properties of whey protein [19,47] [48]. As shown in Figure 3B, the plasma antioxidant capacity of mice after gavage with PWPC was also slightly improved to a degree that may or may not be consistent with an additive effect. 

### 2.3. Toxicity Evaluation of PWPC-GSH System

#### 2.3.1. Clinical Observations, Body Weight, and Food Consumption

During the experiment, there was no observed adverse effects in the experimental groups compared with the control groups. Body weight of all rats increased gradually as the treatment period progressed (Figure 4A). There was no statistically significant difference in body weight between female groups (*p* < 0.05). For male groups, from 16 days, the weight of rats in 1% male group was significantly different from that of the control (*p* < 0.05). However, the body weight changes were observed only in male groups and there was no does-dependent effect, so it was interpreted as having no toxicological significance. During the study, the weight of rats in the experimental groups was comparable to that of rats in the control groups.

Results for food consumption of rats for 28 days are shown in Figure 4B. There was no PWPC-GSH-related toxicity effect observed in experimental groups although there was some significant difference between experimental groups and control groups at some time points. In female rats, there was a statistically significant difference in food consumption of the 0.5% and 4% groups and the control group on the 8th day (*p* < 0.05). However, this phenomenon was only observed in one time point and later returned to normal. Similar phenomenon was also observed in the experiment of Bauter et al. [49], and it was interpreted as having no toxicological relevance and no side effects. In contrast, no change in the food consumption was observed in the male groups.

#### 2.3.2. Hematology

Results of the hematology in all groups were shown in Appendix A. Some statistically significant differences occurred between control and treatment groups (*p* < 0.05). 

In 0.5% and 4% groups, the values of mean platelet volume (WPV) were lower than that of the control group. In the 0.5% group, the lymphocyte (LYM) was significantly different from that of the control group (*p* < 0.05). However, all changes were observed only in male rats. In females, compared with the control, red blood cells (RBC) in the 1% group showed a significantly higher value (*p* < 0.05). This change may be caused by a lack of water and should not be considered as test-substance related. Hemoglobin (HGB) in 1% and 4% female groups was significantly higher than that in the control group (*p* < 0.05). On routine blood tests, there were no toxic reactions associated with taking PWPC-GSH. Although there were significant differences between some of the data and the control group, they were within the historical reference range [50]. These differences can only be observed in one sex, and there is no dose–effect relationship. Therefore, it is considered that this may be due to environmental temperature, placement time and other factors, which cause slight differences.

#### 2.3.3. Serum Biochemistry

Results of serum biochemistry for rats in all groups are shown in Appendix A. Compared to the control group, the 4% male group displayed significantly lower mean values of albumin (ALB) (33.01 ± 1.34 g/L, *p* < 0.05); the changes in ALB value correlate with liver function The values of amylase (AMY), calcium (Ca) and glucose (GLU) were different from those of the control group. Amylase hydrolyzes starch and sugars are originally digested from polysaccharide compounds in foods. The decrease in amylase occurred in diabetes mellitus and severe liver disease. Further examination of rat liver tissue sections showed that there was no lesion in rat liver tissue. The slight decrease in the value of Glu was caused by fasting and starvation in the mice, while the low value of Ca was not clinically significant. These values were comparable to the control and within the bounds of historical data [51]. Compared with the control group, the alanine aminotransferase (ALT) and aminotransferase (AST) values were significantly lower in experimental groups (*p* < 0.05). Previous studies showed that the increased ALT and AST values had clinical significance, while the decreased values may be due to the benefits of PWPC-GSH in inhibiting liver injury [52,53], so it was regarded as not toxicologically relevant; in fact, this finding further illustrates the positive impact GSH has on liver function and protection. 

Amylase (AMY) and GLU in the 1% and 4% female groups were significantly lower than those in the control group (*p* < 0.05) and the inorganic phosphorus (IP) level in the 0.5%, 1% and 4% female groups was significantly higher than that of the control group (*p* < 0.05). Amylase hydrolyzes starch and sugars were originally digested from polysaccharide compounds in foods. The decrease in amylase occurred in diabetes mellitus and severe liver disease. Further examination of rat liver tissue sections showed that there was no lesion in rat liver tissue. A slight decrease in Glu is usually caused by starvation. High IP values have no clinical significance. For males and females, the creatine kinase (CK) values in the control groups were not within the reference range; the difference was due to instrumental error. The difference in CK was interpreted as having no toxicological significance if there was no associated microscopic change in the heart. 

In blood biochemical tests, although there were significant differences in some data compared with the control group, there was no correlation between these changes in the organism and the dose of chemical exposure, so the effect was considered to be independent of the exposure [49].

#### 2.3.4. Relative Organ Weights and Histopathology

Relative organ weight can reflect the nutritional status and damage of the animal internal organs. As shown in Table 1, male rats of experiment groups showed significantly lower final body weights than the control group (*p* < 0.05). In male rats, compared with the control group, there was no significant difference in the relative organ weight of rats in all groups except liver and kidney in the 4% group (*p* < 0.05). There was no evidence of liver and kidney lesions in the hematological analysis. Further observation of the liver and kidney by necropsy and histopathological tests confirmed that there was no damage or lesion in the liver and kidney. For female rats, the 4% group showed significantly lower final body weight (*p* < 0.05). There was no significant difference in relative organ weights between female rats in the experiment groups and the control group (*p* < 0.05). 

At autopsy, no treatment-related gross pathological changes were observed in any group of rats treated with PWPC-GSH, so histopathological examinations were performed on animals in the control and high-dose groups of both sexes. As shown in Figure 5, histopathological findings observed consisted of degeneration of some pyramidal cells in the brain and the c; local slight dilatation of myocardial fiber tract space; focal inflammatory cell infiltration in the liver; renal tubular epithelial cell edema; slight degeneration of the testicular tubules. These lesions are very mild and mostly spontaneous in animals used in the experiment [43,49]. Therefore, these results were independent of the use of PWPC-GSH which could be observed in the age and strain of the rats used in this study.

## 3. Materials and Methods

### 3.1. Materials

Whey protein concentrate (WPC) was provided by Fonterra Co-operative Group (Auckland, New Zealand). GSH at purity of 98.7% was obtained from Kyowa Hakko Bio Co., Ltd (Tokyo, Japan). The GSH powder was stored in the dark to avoid oxidation. A reduced glutathione assay kit (A006-2-1), total antioxidative capacity measurement kit (ABTS method) (A015-2-1) were purchased from Nanjing Jiancheng bioengineering institute (Nanjing, China). Pentobarbital sodium, formalin and absolute ethanol were provided by Beijing chemical Works (Beijing, China).

### 3.2. Preparation of Polymerized Whey Protein Concentrate Based GSH Delivery System

The preparation of polymerized whey protein concentrate (PWPC) solution was performed according to Khan’s method [54]. WPC solution (8%, *w*/*v*) was prepared by dissolving whey protein concentrate powder in deionized water. The stock solution was stored at 4 °C overnight for complete hydration. WPC solution was returned to ambient temperature and the pH was adjusted to 7 followed by thermal treatment at 80 °C for 15 min in a water bath. PWPC solution was obtained by cooling heated WPC solution in mixed water–ice quickly to room temperature (25 ± 1 °C). PWPC-GSH was obtained by mixing PWPC solution with GSH powder at a weight ratio of 1:1.25 and then stirring for 30 min for complete interaction.

### 3.3. Characterization of PWPC-GSH Delivery System 

#### 3.3.1. Particle Size and Zeta Potential Measurement

The PWPC and PWPC-GSH solutions were slowly added into sample pool until shading was 10–15% and the particle size was measured using Mastersizer 3000 (Malvern Instruments Ltd., Worcestershire, UK). Zeta potential was determined using a Malvern Nano Zetasizer (Malvern Instruments Ltd., Worcestershire, UK).

#### 3.3.2. Flow Behavior Measurement

Flow behavior of suspensions was measured by a Haake Mars 40 Rheometer (Thermo Scientific, Waltham, MA, USA) equipped with a parallel plate which is 60 mm in diameter and 0.1 mm in thickness. The measurement gap width was set at 1000 μm and shear rates ranged from 0.1 to 300 s^−1^.

#### 3.3.3. Differential Scanning Calorimetry (DSC) Measurement 

Thermal characteristics of samples were analyzed using Differential Scanning Calorimeter (DSC3, Mettler Toledo, Switzerland). Sample suspensions were pre-frozen overnight at −80 °C and then dried at 4 °C and 0.3 MPa for 24 h. A small amount of sample (3 mg) was placed and sealed in an aluminum plate with empty pan taken as the control. Calorimetric measurement was performed with the temperature increased from 20 to 220 °C at a heating rate of 10 °C/min.

#### 3.3.4. Fourier Transform Infrared Spectrometry (FT-IR) Measurement

FT-IR spectra of freeze-dried samples were recorded using a Fourier Infrared Spectrometer (NICOLET IS10, Thermo Scientific, Waltham, MA, USA) in the wavelength range of 500–4500 cm^−1^. 

#### 3.3.5. Transmission Electron Microscopy (TEM) Measurement

Appropriate amount of dispersion diluted to 0.02 mg/mL was dropped onto copper grid and the excess sample was removed with filter paper. Uranyl acetate solution (2%, *w*/*v*) was added and staining were kept for about 15 s. After drying at room temperature for 60 min, micromorphology photos were taken using a Transmission Electron Microscopy (H-7650, HITACHI, Tokyo, Japan) at 100 kV with magnification of 12,000.

### 3.4. Animal Care

Institute of Cancer Research (ICR) mice (male, 3 weeks, 18–22 g) and Sprague Dawley (SD) rats (male and female, 3 weeks) at Specific Pathogen Free (SPF) grade were provided by Beijing HFK Bioscience Co., Ltd. (Beijing, China). All animals were housed in plastic animal cages in a ventilated room. The room was maintained at 20–26 °C and 40–60% relative humidity with a 12-h light/dark cycle. Water and commercial laboratory complete food were available ad libitum. They were acclimated to the environment for 7 days before the experiment. 

### 3.5. In Vivo Pharmacokinetic Study and Antioxidant Activity Analysis

Blood samples of mice were collected after oral administration of normal saline (control), PWPC, GSH, and PWPC-GSH solutions by gavage at several time points (0, 15 min, 30 min, 1 h, 2 h, and 4 h) in each group of 6 mice. According to the previous study [32], the dose of PWPC-GSH was adjusted to an equivalent amount of 100 mg/kg of GSH. Blood samples (about 0.5 mL) were collected in 1.5-mL centrifugal tubes with 30 μL heparin solution by removing one eye of the mice. Plasma samples were obtained by centrifugation of blood at 6000 rpm for 2 min at room temperature and GSH concentration was determined by an assay kit. Pharmacokinetic parameters including maximum concentration (Cmax), time to maximum concentration (Tmax), area under the concentration–time curve (AUC), half-life (T1/2), and mean residence time (MRT) were estimated by DAS 2.0 (BioGuider Co., Shanghai, China). Antioxidant activity of plasma was also determined using the total antioxidative capacity (T-AOC) measurement kit (ABTS method). The total antioxidant capacity of Trolox was determined by using Trolox as the standard, and the antioxidant Capacity of the sample can be expressed by Trolox-equivalent Antioxidant Capacity (TEAC).

### 3.6. Sub-Chronic Toxicity Evaluation

#### 3.6.1. Experimental Design

In the 28-day subchronic toxicity test, 80 Sprague Dawley (SD) rats (40 males and 40 females) with a body weight of about 75g were divided into 4 groups (10 rats/sex/group). Animals were given a formulated diet containing freeze dried PWPC-GSH powder in daily dose of 0% (*w*/*w*, control), 0.5% (*w*/*w*, low dose), 1% (*w*/*w*, medium dose), and 4% (*w*/*w*, high dose) which corresponds to 0.25%, 0.5% and 2% percentage for GSH. The PWPC-GSH-containing diets were manufactured by Beijing HFK Bioscience Co., Ltd. (Beijing, China). 

#### 3.6.2. Clinical Examination, Body Weight, and Food Consumption

Coat condition, skin, mucous membranes, secretions, excretions, autonomic nervous system activity, changes in gait, and posture of each rat were observed each day throughout the study. Body weights at interval of 4 days were weighed and recorded. Food intake measured and expressed as mean food consumption (expressed as g/rat/day) was calculated for the corresponding intervals. Final weights (fasting) were recorded prior to the scheduled autopsy for calculating the relative weight of organs.

#### 3.6.3. Clinical Pathology

At termination, all the animals were fasted for 12 h, but were free to drink water. Rats were anesthetized with 2% sodium pentobarbital solution of 0.2 mL/100 g weight [55]. Hematological and serum chemistry examination was performed by cardiac blood collection. In hematology analysis, ethylenediamine tetraacetic acid dipotassium salt (EDTA-2K) was used as an anticoagulant and was performed using the Exigo Animal Hematology Analyzer (Exigo-EOS, Sweden).In the serum chemistry examination, the blood samples were centrifuged at room temperature at 10,000 rpm for 3 min to obtain serum, and the serum chemical analysis was performed on an automatic biochemical analyzer (SMT-120V, Chengdu, China).

#### 3.6.4. Organ Weights and Histopathology

At termination, all rats were anaesthetized by pentobarbital sodium. Then, a complete gross pathology examination was conducted by visual inspection. In the process of necropsy, the brain, heart, lung, liver, spleen, kidney and bladder of all animals were taken, and the reproductive organs including ovaries and uterus of female rats were taken, and the testis, epididymis and seminal vesicle of male rats were taken. All organs were removed and weighed. The organ coefficient was calculated by the ratio of organ weight to body weight, expressed as %. Tissue sections from these organs were fixed with 10% buffered formaldehyde except testes were fixed in Bouin solution, embedded in paraffin, sectioned at 2–5 μm, mounted on glass microscope slides, stained with standard hematoxylin–eosin and examined using light microscopy. All histopathology procedures were carried out in the College of Animal Science and Veterinary Medicine, Jilin University.

### 3.7. Statistical Analysis

The significance differences of quantitative data between control and experimental groups were calculated using Version SPSS 20 (SPSS Inc., Chicago, IL, USA). The data were expressed as means ± standard deviation. Levene’s test was used to conduct homogeneous analysis of the data. When the data were homogeneous, the LSD method was used for further analysis; when the data were heterogeneous, Dunnett’s test was used for analysis. All statistical tests were performed at the *p* < 0.05 and *p* < 0.01 levels of significance.

## 4. Conclusions

In this study, a novel polymerized whey protein concentrate-based glutathione oral delivery system with high stability was successfully developed resulting in greater delivery efficiency as evidenced by in vivo pharmacokinetic data and antioxidant activity analysis. Oral administration of PWPC-GSH in diet concentration up to 4% (*w*/*w*) for 28 days had no adverse effects on male and female rats. All results suggested that thermal treatment induced polymerized whey protein concentrate represents a viable, effective potential delivery system with the ability to enhance oral bioavailability of GSH. 

## Figures and Tables

**Figure 1 molecules-26-01824-f001:**
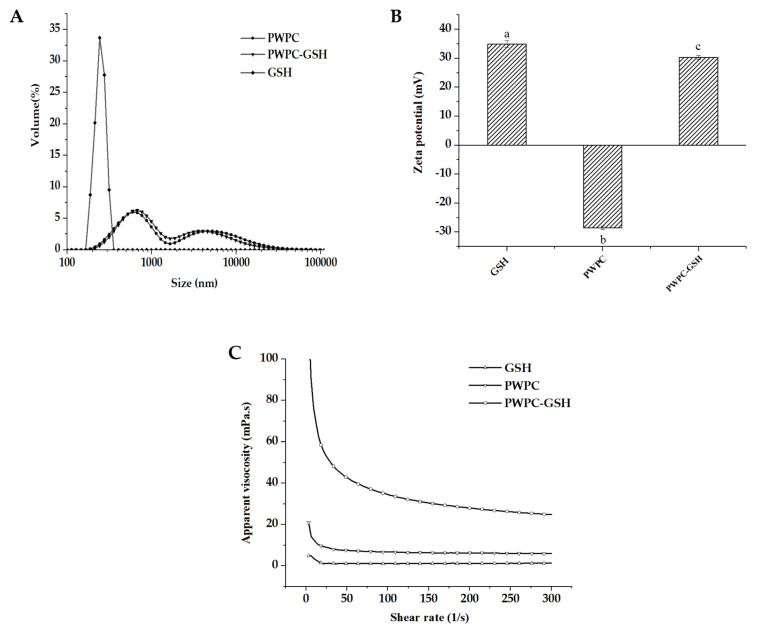
(**A**) Particle size distribution, (**B**) zeta potential, (**C**) apparent viscosity of glutathione (GSH), polymerized whey protein concentrate (PWPC) and polymerized whey protein concentrate encapsulated GSH (PWPC-GSH). In Figure 1B, completely different lower-case letters between samples mean significant difference (*p* < 0.05).

**Figure 2 molecules-26-01824-f002:**
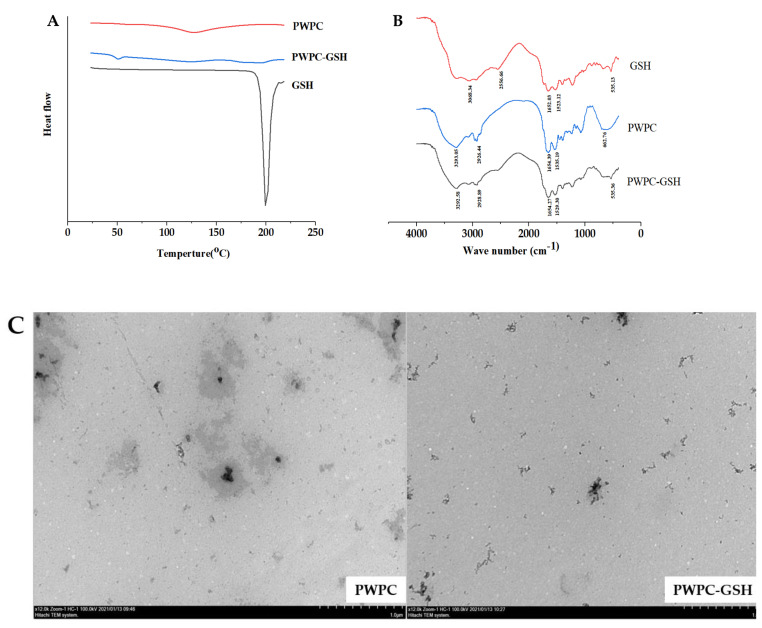
(**A**) Differential scanning calorimetry (DSC), (**B**) FT-IR spectrum, (**C**) TEM image of PWPC and PWPC-GSH.

**Figure 3 molecules-26-01824-f003:**
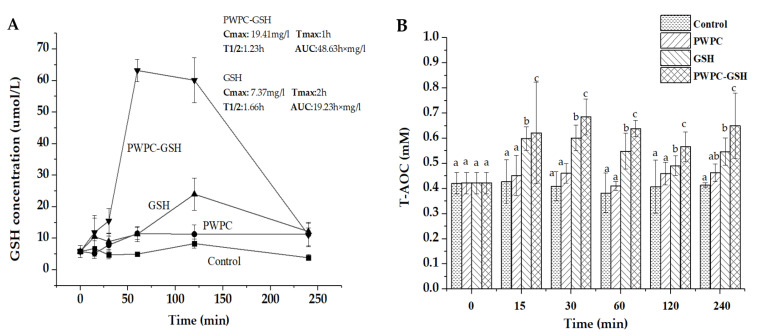
(**A**) GSH concentration (μM) in plasma of mice after oral administration of normal saline (control), PWPC, GSH, and PWPC-GSH. (**B**) In vivo antioxidant activity of plasma of mice after oral administration of normal saline (control), PWPC, GSH, and PWPC-GSH measured by an assay kit (ABTS method). In Figure 3B, completely different letter between samples at the same timepoint means significant difference (*p* < 0.05).

**Figure 4 molecules-26-01824-f004:**
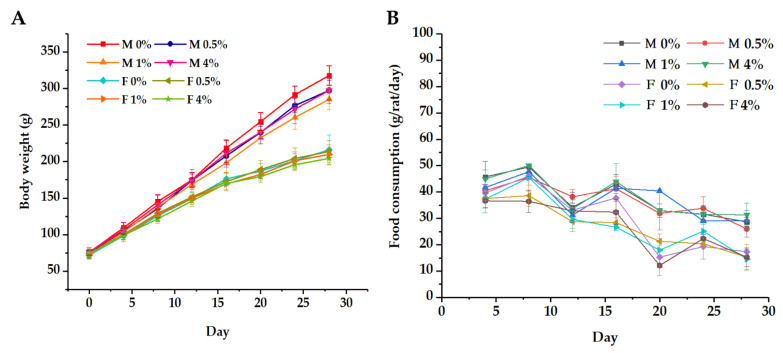
(**A**) Body weight and (**B**) food consumption curves of rats in a 28-day feeding test, 0%, 0.5%, 2% and 4% represent daily intake of 0% (*w*/*w*, control), 0.5% (*w*/*w*, low dose), 1% (*w*/*w*, medium dose) and 4% (*w*/*w*, high dose) of freeze-dried PWPC-GSH powder, respectively.

**Figure 5 molecules-26-01824-f005:**
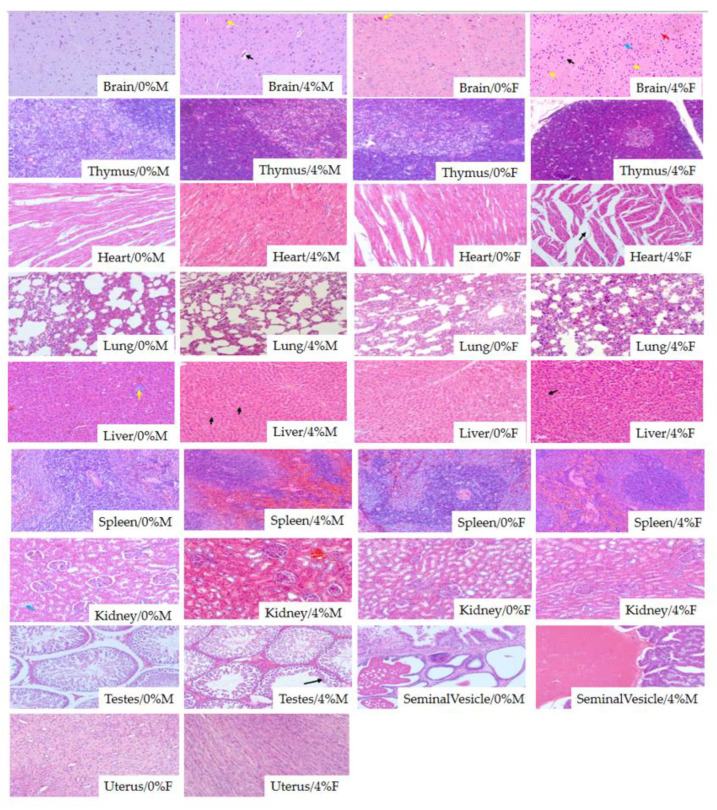
Microscopic examination results of organs for rats in the control and 4% groups. The values 0%, 4% represent daily intake of 0% (*w*/*w*, control), 4% (*w*/*w*, high dose) of freeze-dried PWPC-GSH powder. M and F represent male and female rats, respectively. For pictures of brain, black arrows represent vascular endothelium, yellow arrows represent the nucleus, red arrow are neurons and blue arrow are pyramidal cells. For picture of heart, black arrow represents the gap of cardiac muscle fiber bundle. For picture of liver, black arrows represent prototype vacuoles with regular shapes and yellow arrow represent red blood cells. For picture of testes, black arrows represent sperm cells.

**Table 1 molecules-26-01824-t001:** Relative organ weights for male rat.

		0%	0.5%	1%	4%
Body weight	M	296.8 ± 11.47	272.78 ± 10.28 **	261 ± 12.59 **	275 ± 18.70 *
	F	203.80 ± 10.14	205.20 ± 16.01	199.70 ± 11.93	192.10 ± 8.77 *
Brain	M	5.35 ± 0.85	5.90 ± 2.11	6.45 ± 0.63 *	6.67 ± 0.67 *
	F	7.14 ± 1.40	8.52 ± 0.86	8.22 ± 1.02	8.65 ± 0.84
Thymus	M	2.58 ± 0.60	3.01 ± 0.42	2.37 ± 0.38	2.66 ± 0.47
	F	3.36 ± 0.46	3.31 ± 0.38	3.40 ± 0.52	3.31 ± 0.44
Heart	M	3.80 ± 0.30	4.03 ± 0.50	3.92 ± 0.28	3.89 ± 0.28
	F	4.10±0.01	3.99 ± 0.26	4.26 ± 0.37	4.08 ± 0.31
Lung	M	4.74 ± 0.38	5.31 ± 1.08	4.98 ± 0.52	4.78 ± 0.31
	F	5.44 ± 0.55	5.09 ± 0.39	5.19 ± 0.35	5.13 ± 0.33
Liver	M	39.00 ± 4.33	37.22 ± 4.83	36.62 ± 2.96	34.92 ± 2.83 *
	F	35.76 ± 2.74	38.28 ± 7.27	34.54 ± 2.14	36.08 ± 1.55
Spleen	M	2.79 ± 0.92	2.78 ± 0.72	2.47 ± 0.35	2.79 ± 0.39
	F	2.51 ± 0.32	2.56 ± 0.40	2.32 ± 0.34	2.57 ± 0.21
Kidney	M	9.16 ± 0.70	9.32 ± 0.39	9.38 ± 0.38	9.66 ± 0.51 *
	F	8.83 ± 0.46	8.54 ± 1.53	8.70 ± 0.31	9.47 ± 0.51
Bladder	M	0.28 ± 0.04	0.31 ± 0.07	0.32 ± 0.05	0.29 ± 0.07
	F	0.35 ± 0.03	0.36 ± 0.07	0.35 ± 0.04	0.35 ± 0.06
Testes	M	6.796 ± 1.79	8.62 ± 1.05	8.25 ± 0.23	8.67 ± 0.76
Epididymis	M	0.57 ± 0.11	0.48 ± 0.15	0.58 ± 0.09	0.59 ± 0.05
Seminal Vesicle	M	1.95 ± 0.58	1.47 ± 0.42	1.11 ± 0.68	1.59 ± 0.50
Ovary	F	0.71 ± 023	1.65 ± 0.28	0.54 ± 0.16	0.56 ± 0.14
Uterus	F	2.11 ± 0.62	1.98 ± 0.62	1.75 ± 0.65	1.99 ± 0.98

Note: * means the significant level is 0.05, ** means significant level is 0.01 compared with the control group. The values 0%, 0.5%, 2% and 4% represent daily intake of 0% (*w*/*w*, control), 0.5% (*w*/*w*, low dose), 1% (*w*/*w*, medium dose) and 4% (*w*/*w*, high dose) of freeze-dried PWPC-GSH powder; M and F represent male and female rats, respectively.

## Data Availability

Date available in a publicly accessible repository.

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
