# Peer review of "Polymerized Whey Protein Concentrate-Based Glutathione Delivery System: Physicochemical Characterization, Bioavailability and Sub-Chronic Toxicity Evaluation"

_molecules, 2021, doi:10.3390/molecules26071824_

Round 1

Reviewer 1 Report

In this manuscript, the authors for the first time report that polymerized whey protein concentrate may be an effective carrier of GSH thereby achieving its improved bioavailability and antioxidant capacity. Since the paper could be of interest for the scientific community, I'd recommend publishing it in "Molecules" after some minor revisions of the manuscript:

  1. On the y axis of Figure 3A change the word „concentrationg“ to „concentration“
  2. On the y axis of Figure 3B is the abbreviation T-AOC with no unit specified. Also, for the abbreviation T-AOC, the full name should be written in the text of the manuscript because it only appears in this place.
  3. The results in Figures 4A and 4B are not clear enough so I suggest the authors to improve them.
  4. No unit for y axis name (Food consumption) in Figure 4B!
  5. According to the results in Table 1, the statement in the sentence (pg.7, line 197-198): "The increase in liver weight and decrease in kidney weight…. " should be corrected to " The decrease in liver weight and increase in kidney weight… ".
  6. On pg. 7, for the first time in the text of the manuscript, abbreviations such as LYM, HGB, PLT, etc. appear. Abbreviations should be defined in parentheses the first time they appear in the main text.

Reviewer 2 Report

The study by Zhang et al. demonstrated the safety and functional characteristics of a novel encapsulation and delivery system of polymerized whey protein concentrate and GSH, and found that the polymerized concentrate could be a viable delivery system with the tendency to enhance oral bioavailability of GSH and poses no toxicological threat within the limit of the investigated doses and experimental period of 28 days. The following issues need to be considered:

  1. The inability of the authors to give justifiable scientific reason(s) for the observed effects shown in most of the experiments performed, especially on the hematological parameters is a concern that cannot be overlooked. For instance, the authors stated that the observed significant difference in PLT for the 4% group relative to the control group was historical. Again, a similar observation was made with RBC and HGB in the 1% group and the authors considered these changes to lack of water and 'incidental', respectively (Lines 159 - 168). Such conclusions do not hold grounds in science and it is highly recommended that the affected portions of the manuscript be revised accordingly.
  2. The authors made statements such as '.....significantly higher... (Line 124)...,.....no statistically significant difference.....(Lines 138 -140)...etc' bothering on DATA ANALYSIS using statistical operations/applications without a clear section/statement in the manuscript showing what kind of statistical analyses were performed. This was further compounded by the use of 'error bars' on Figures 3 and 4 and asterisk (*) in Table 1. It is hence suggested that the type of statistical analysis used in the study be included and comprehensively detailed to ascertain the true level of significant variations in some of the data obtained in the study.
  3. Explain how the doses and dosage of the test agents (PWPC, GSH and PWPC-GSH) for the pharmacokinetic study were arrived at and the rationale for the choice(s). This is equally applicable to the sub-chronic toxicity evaluation, where the guideline used was not stated. It is also important to provide relevant reference(s) adopted for these studies.
  4.  Please, provide information on the weight of rats used as you have done for the mice and include a suitable reference for the anaesthetic method used in the study.
  5. Lines 313 - 315: Revise the content to be specific on the sex of the animals used.    
  6. Please, provide information on the source, purity and stability of the GSH used.
  7. How were the relative organ weights determined and in what unit have they been expressed? Also include the 'unit' for the 'Food consumption' in Figure 4.
  8. Kindly revise the entire manuscript for grammatical (e.g section 2.6.3), spelling (e.g. reseach [Line 73]) and syntax errors to improve the overall editorial quality. 

Round 2

Reviewer 2 Report

The Authors have made effort to improve on the quality of the work but the following minor concerns still need to be addressed:

  1. Despite that it was claimed that justifiable reasons have been given for most of the results obtained, it is still surprising to see things like '..which was thought to occur by chance' (Pg 6, Line 153), 'were considered incidental...(Pg 7, Line 209') etc. These kind of inferences are scientifically weak and must be amended.
  2. Provide suitable reference(s) adopted for sections 2.2 (Lines 240 - 247), 2.5 (Lines 286 - 289), and 2.6.3 (Lines 317 - 318).
  3. Again, the manuscript must be further revised for improved editorial quality bothering on grammar and syntax errors. e.g. revise the designations used for the sub-sections in SECTION 3, particularly from 2.2 to 2.7, which is expected to be 3.2, 3.3 etc to 3.7 accordingly. Kindly check and revise. Also, '...rats were shown in Table S2' (Line 172), '..This results were....' (Line 104) etc all needs to be adequately revised.
  4. Please, include the information on the source, purity and stability of the GSH used in section 3.1 of the manuscript. 
